# A Quick reCAP: Discovering *Cryptococcus neoformans* Capsule Mutants

**DOI:** 10.3390/jof10020114

**Published:** 2024-01-30

**Authors:** Daphne Boodwa-Ko, Tamara L. Doering

**Affiliations:** Department of Molecular Microbiology, Washington University in St. Louis, St. Louis, MO 63110, USA; daphne.k@wustl.edu

**Keywords:** capsule, *Cryptococcus neoformans*, genetic screen, mutant library

## Abstract

*Cryptococcus neoformans* is an opportunistic fungal pathogen that can cause severe meningoencephalitis in immunocompromised hosts and is a leading cause of death in HIV/AIDS patients. This pathogenic yeast is surrounded by a polysaccharide capsule that is critical for virulence and plays important roles in host-pathogen interactions. Understanding capsule biosynthesis is therefore key to defining the biology of *C. neoformans* and potentially discovering novel therapeutic targets. By exploiting methods to identify mutants deficient in capsule, June Kwon-Chung and other investigators have discovered numerous genes involved in capsule biosynthesis and regulation. Successful approaches have incorporated combinations of techniques including mutagenesis and systematic gene deletion; complementation and genetic screens; morphological examination, physical separation, and antibody binding; and computational modeling based on gene expression analysis. In this review, we discuss these methods and how they have been used to identify capsule mutants.

## 1. Introduction

*Cryptococcus neoformans* is an opportunistic fungal pathogen that is found ubiquitously in the environment, primarily associated with soil and pigeon guano. Inhalation of *C. neoformans* generally has little impact on immunocompetent hosts, although it can cause latent infections. In immunocompromised hosts, however, *C. neoformans* may become reactivated from prior exposure or establish a new pulmonary infection and subsequently disseminate to other tissues. If it enters the brain, *C. neoformans* can cause an often-fatal meningoencephalitis. As of 2023, there were an estimated 194,000 cases of cryptococcal meningoencephalitis per year globally, leading to 147,000 deaths [1].

*C. neoformans* is known for its characteristic polysaccharide capsule, which surrounds the cell wall and becomes dramatically enlarged during host infection. This structure inhibits phagocytosis by host macrophages [2,3] and helps engulfed cryptococci survive intracellularly by protecting against reactive oxygen species and free radicals within the phagosome [4]. Capsule polysaccharides are also shed into the host milieu, where they modulate the host immune response to promote fungal survival [5,6]. Consistent with these significant effects on pathogenic processes, absence of capsule causes avirulence in a mouse model of infection [7], thus demonstrating that this structure is essential for disease.

The polysaccharide capsule is associated with the cell wall and is made up of two repeating polysaccharides: glucuronoxylomannan (GXM) and glucuronoxylomannogalactan (GXMGal). GXM comprises approximately 90% of the capsule mass whereas GXMGal constitutes the remaining 10% [8]. As its name implies, GXM is composed of a mannose backbone with variable glucuronic acid and xylose substitutions. GXMGal consists of a galactan backbone with galactomannan side chains bearing glucuronic acid and xylose. Mannose residues of both polymers may also be *O*-acetylated [9].

Early on, investigators realized that elucidating capsule biosynthesis would be important in understanding the biology of *C. neoformans* and in potentially identifying therapeutic targets. To advance this goal, several groups isolated mutants that were deficient in capsule [10,11,12]. This led to efforts to determine the corresponding genes, which were pioneered by June Kwon-Chung, working primarily with Yun Chang [7,13,14,15]. They identified several genes required for capsule formation by complementing acapsular mutants, experiments that fulfilled Koch’s molecular postulates [16] and validated capsule as an essential virulence factor in *C. neoformans*. These seminal studies launched a new area of cryptococcal research dedicated to elucidating capsule synthesis and regulation at the molecular level. Below, we briefly review several key approaches that contributed to the discovery of capsule mutants and enabled the growth of this field.

## 2. Methods for Assessing Capsule Mutants

To study capsule mutants, they must first be differentiated from cells with normal capsules. Defects in capsule synthesis lead to various phenotypic changes that can be used to screen and assess capsule mutants. Some physical properties may be monitored at the population level. For example, acapsular strains sediment poorly from liquid medium and are less voluminous than wild-type strains. Acapsular colonies grown on solid medium also look dull, dry, and clumpy, in contrast to wild-type colonies which appear shiny and smooth.

Changes in capsule thickness at the cellular level can be readily assessed by negative staining with India ink, usually after incubation in capsule-inducing conditions to make visualization easier (Figure 1, top row). This method is most suitable for viewing dramatic changes in capsule thickness. Another way to assess capsule thickness, which is potentially more sensitive, is to visualize its outer edge with anti-capsule antibodies and its inner boundary with cell wall stains (Figure 1, bottom row).

Anti-capsule antibodies of varying specificity may also be used to characterize alterations in capsule structure, such as density, modification, or monosaccharide composition. For example, one study using a panel of anti-capsule antibodies found that different serotypes of *C. neoformans*, which have varied capsule composition, have different patterns of antibody reactivity [17]. Binding of some anti-capsule antibodies has also been shown to be sensitive to *O-*acetylation [18].

For detailed structural analysis, chemical techniques can be used to compare capsule polysaccharides between mutant and wild-type cells. For these studies, shed capsule material is typically recovered from conditioned media for determination of monosaccharide composition and linkage [19,20].

Capsule plays a significant role in host-pathogen interactions, particularly in modulating phagocytosis; acapsular and hypocapsular cryptococci are more readily engulfed than wild type [2,3]. Assessing these interactions, for example by measuring phagocytic uptake, can also be used to identify potential capsule mutants [21,22].

All of the methods described in this section have been used to identify and characterize individual capsule mutants. Many of them have also been utilized in larger scale efforts to identify mutants of interest, as will be described below.

## 3. First Cloning of ‘Capsule Genes’

In the 1990s, Kwon-Chung and Chang first identified several genes necessary for normal capsule synthesis. The concept was straightforward: transform an acapsular strain with wild-type genomic DNA and determine the gene or genes necessary to complement the defect. Using this general approach, they identified four genes that are essential for capsule synthesis [7,13,14,15,19]. For each of them, they began with a strain from a population subjected to mutagenesis (via 5-fluoroorotic acid treatment, UV irradiation, or *N*-methyl-*N’*-nitro-*N*-nitrosoguanidine treatment) that exhibited dry colony morphology and had been confirmed to be acapsular by microscopy. They then transformed the mutant with a genomic DNA library (from J. C. Edman [23] or their own group) and isolated transformants with restored capsule. They did this using a two-polymer system initially developed by Kozel and Cazin [24], in which encapsulated and acapsular cells partition into polyethylene glycol (PEG) 8000-rich and dextran-rich phases, respectively (Figure 2). Next, they recovered the plasmid responsible for complementing the acapsular strain and determined the minimal DNA sequence required for complementation. They showed that deleting the corresponding genes (*CAP59*, *CAP60*, *CAP64*, and *CAP10*) from wild-type *C. neoformans* yielded acapsular strains and used these deletion strains to show that these sequences influence phenotypes including capsule production, virulence, complement activation and binding, and host immune response. Sequence homology suggests that these genes encode glycosyltransferases [25,26], although the biochemical functions of their products remain to be determined.

Importantly, these landmark studies fulfilled molecular Koch’s postulates—deletion of the *CAP* genes led to loss of capsule and decreased virulence, which were restored upon gene complementation. Through these studies, Kwon-Chung and colleagues proved that the capsule of *C. neoformans* is a key virulence factor and that specific genes are responsible for capsule phenotypes, thus bringing the field of cryptococcal capsule biology into the molecular era. In the almost twenty years since these elegant studies, various approaches have been developed to further identify capsule mutants and define their function. Below, we will discuss a few of these techniques and their applications.

## 4. Screening with Anti-Capsule Antibodies

As discussed above, capsule mutants with gross changes in capsule thickness as determined by colony morphology, density, or negative staining, had been isolated and studied by multiple groups. In a novel screening approach reported in 2001, Janbon and colleagues used anti-capsule antibodies to identify capsule structure mutants (*CAS* mutants). Their rationale was that even if their specificity was unknown, capsule antibodies could potentially differentiate between wild-type cells and those with structurally altered capsules.

For these studies, the investigators first UV-irradiated the *C. neoformans* wild-type strain JEC21 to induce mutations and then used an anti-capsule monoclonal antibody, CRND-8, to screen for non-reactive transformants [27]. Similar to earlier work by Kwon-Chung and colleagues, Janbon et al. next transformed one such mutant with a genomic library (provided by B. Wickes [28]) and identified transformants that recovered the capacity to bind CRND-8. They established the sequence responsible for complementation, which they named *CAS1*, and noted that it encodes a putative glycosyltransferase. NMR analysis of purified GXM from *cas1*Δ showed complete loss of GXM *O-*acetylation, confirming a role for this gene in determining capsule structure.

In a second study the following year, the same group used anti-capsule antibodies to identify 23 more mutants with capsule that differed from that of wild-type cells, in this case using JEC43 as the parental strain. These strains were tested against a panel of five monoclonal anti-GXM antibodies (CRND-8, 4H3, 5E4, 2H1, and E1) and classified into six groups (*CAS1*–*CAS6*) according to antibody reactivity [29]. JEC43 binds the first four antibodies but not E1 [29]. Interestingly, they found that mutants lacking *CAS2* (also termed *UXS1*) only bind 4H3 of these five antibodies. Mutants lacking *CAS3* gain E1 binding and lose CRND-8 binding compared to wild type. Later work showed that Uxs1 catalyzes synthesis of a precursor required for capsule xylosylation [30]; consistent with this finding, *uxs1*Δ GXM lacks xylose residues [29]. GXM from the *cas3*Δ mutant is only partly *O-*acetylated and has more xylose than wild-type GXM [31]. These findings strongly validated this approach to dissecting capsule synthesis.

Another antibody-based screening assay, developed by our group, uses antibodies to demarcate the extent of the capsule, rather than for their specificity. For this method, we stain cells with both anti-capsule monoclonal antibodies and a cell wall stain (as in Figure 1, panel D) in multi-well format. Automated imaging of these samples allows rapid measurement of cell size and capsule thickness of many individual cryptococci, data which can be used to assess populations [22]. This technique has been validated with known hypercapsular, hypocapsular, and wild-type strains [32] and has been used to screen for capsule mutants [33].

## 5. Screening Deletion Collections for Capsule Phenotypes

While the original Kwon-Chung studies focused on individual capsule-deficient mutants, subsequent investigations have taken advantage of deletion mutant collections to screen for capsule phenotypes in a higher throughput manner. Here we will highlight the largest deletion collections, which were produced by the Madhani and Bahn groups.

About a decade after the first *CAP* mutant was reported [7], Madhani and colleagues began to develop a deletion mutant collection, with the goal of eventually deleting every non-essential gene in the *C. neoformans* genome. This library, first reported in 2008, has since grown to almost 5000 mutants (out of a total of approximately 6500 genes in *C. neoformans* [34]) and has had a major impact as a community resource. Using high-throughput biolistic transformation, this group initially targeted 1500 genes for replacement with a nourseothricin resistance cassette and unique barcode. They successfully generated 1180 unique deletion mutants [35,36], which they screened for in vivo infectivity in a mouse model, growth at 37 °C, melanin production, and capsule production (by colony morphology). The last assay yielded eleven mutants with dry colony morphology, six of which lacked genes that had already been implicated in capsule synthesis (*CAP59*, *CAP64*, *CAP60*, *CAP10*, *CAS35*, *NRG1*) [35].

Of the five novel genes (*GAT201*, *SSN801*, *HOS2*, *SET302*, and *CPL1*), deletion of *GAT201* led to the greatest virulence defect and total loss of capsule. Liu et al. determined that the encoded transcription factor is a key regulator of virulence and capsule-independent antiphagocytic mechanisms. They also found that Gat201 upregulates the expression of downstream transcription factors that in turn activate capsule-related genes [35].

In a more targeted approach, the Bahn group aimed to elucidate the capsule regulatory network by developing three deletion mutant libraries of strains lacking predicted transcription factors, kinases, or phosphatases [37,38,39]. They found that 49 of 155 putative transcription factors, 33 of 129 putative kinases, and 20 of 114 putative phosphatases impacted capsule production as determined by India ink imaging or packed cell volume. While these hits included factors with known roles in capsule formation, such as Gat201 and adenylyl cyclase, the screens also uncovered novel hits. For example, deletion of *YAP1* led to severely reduced capsule production, as well as enhanced melanin production and reduced virulence. Yap1 is part of a MAP kinase signaling cascade that responds to environmental stresses (including oxidative, osmotic, cell wall, and membrane stressors) and plays a role in antifungal drug resistance and virulence [40,41]. This work during the last decade by Bahn and colleagues has contributed to unraveling capsule regulation networks.

## 6. Modeling Capsule Regulatory Networks

With the availability of genome sequence information and genome-scale experimentation came new methods of identifying capsule genes, including those based on homology and computational modeling. For example, beginning in 2011, Brent and colleagues reported the development of computational models to predict transcription factors involved in the regulation of capsule synthesis [42].

In one approach, these researchers grew cells in a range of growth conditions that stimulate capsule synthesis in order to identify transcriptional signatures associated with capsule enlargement [43]. Their initial studies using RNASeq and ChIPSeq implicated the transcriptional regulator Ada2, which controls histone acetylation in response to stress, in capsule production. Further studies by the Brent group captured a systems-level view of capsule regulation and biosynthesis by analyzing the expression profiles of 41 transcription factor deletion mutants, including 20 that were not previously implicated in cryptococcal virulence. They developed an algorithm (NetProphet) to model regulatory networks based on these expression profiles and another (PhenoProphet) to predict which transcription factors were most likely to regulate capsule production. Validation by gene deletion of predicted capsule-regulating transcription factors supported these predictive models [44]. For example, NetProphet identified Usv101 as a capsule transcriptional regulator and PhenoProphet predicted that it would be required for capsule regulation. Further experiments supported these predictions and allowed delineation of the mechanism: Usv101 was found to directly regulate glycoactive enzymes as well as other transcription factors, including Gat201, Rim101, and Sp1 [45]. Modeling approaches are thus powerful tools for predicting genes that act in capsule synthesis and dissecting the capsule regulatory network.

## 7. Final Thoughts

We have come a long way in the 30 years since June Kwon-Chung and her colleagues identified the first capsule gene and demonstrated that capsule is a virulence factor (Figure 3). We now recognize hundreds of genes that influence capsule [46,47,48] and have defined multiple relevant enzymatic and regulatory pathways. The enormous impact of Dr. Kwon-Chung’s early work on this field has been sustained in the decades since and is likely to extend into the future, as studies continue to reveal the functions of capsule genes and their crucial roles in *C. neoformans* biology.

## Figures and Tables

**Figure 1 jof-10-00114-f001:**
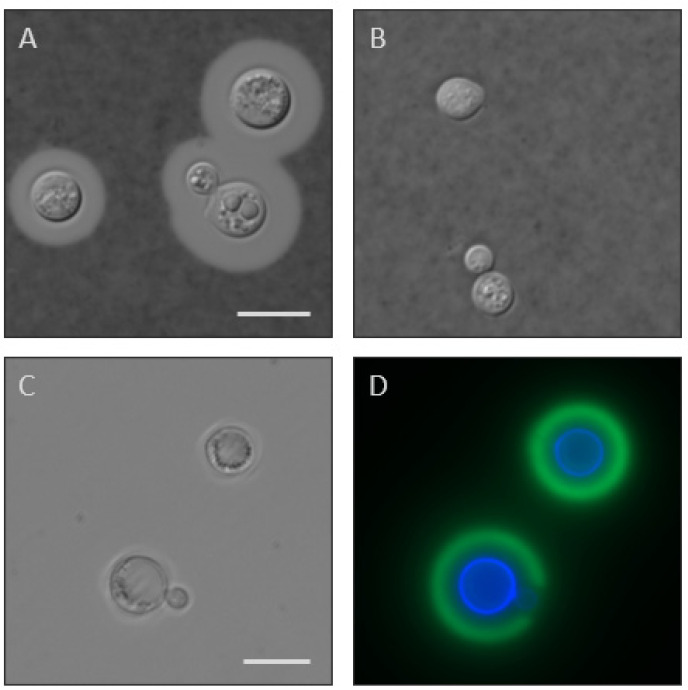
Capsule imaging. Top row, India ink staining of wild-type *C. neoformans* (**A**) and acapsular mutant *cap10*Δ (**B**) grown in DMEM for 24 h. Scale bar, 10 microns. Both images are to the same scale. Bottom row, DIC (**C**) and fluorescent (**D**) micrographs of wild-type *C. neoformans* stained with the cell wall stain calcofluor white (blue) and anti-capsule antibody 302 tagged with AlexaFluor488 (green). The apparent gap in staining of the lower cell reflects the site of new bud growth that occurred after capsule staining. Scale bar, 10 microns. Both images are to the same scale.

**Figure 2 jof-10-00114-f002:**
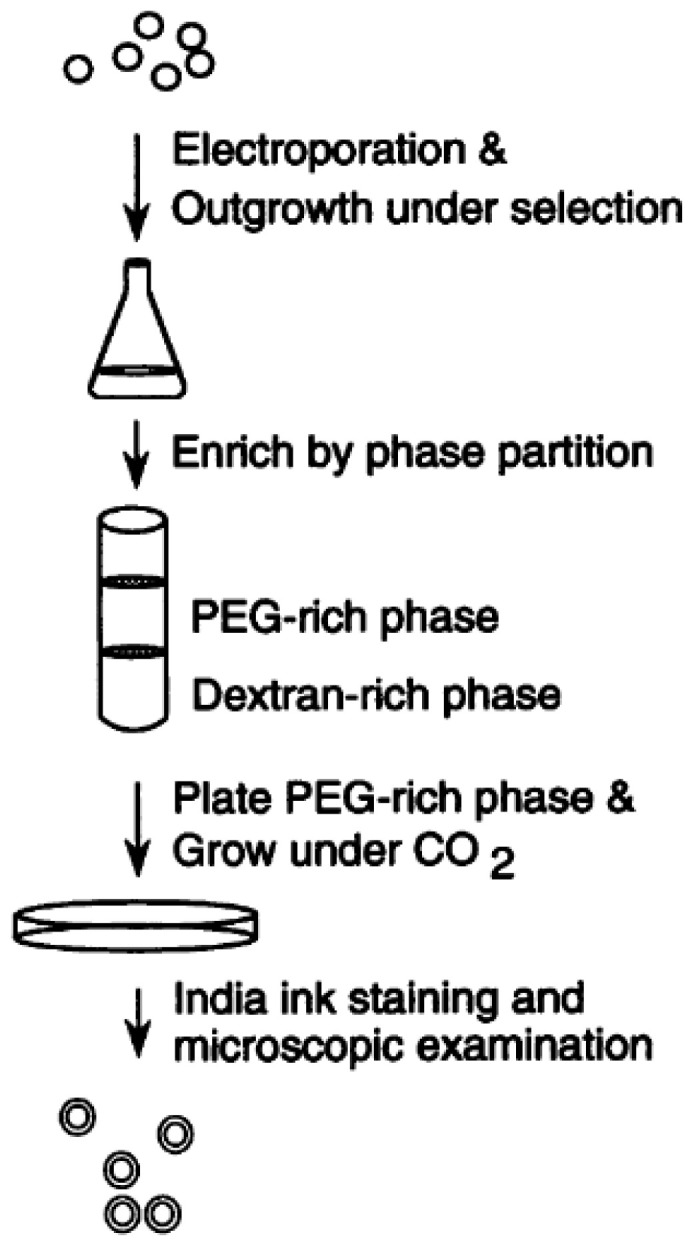
Selection of encapsulated transformants following complementation of acapsular strains. Figure was originally published by Chang and Kwon-Chung in 1994 with Taylor & Francis Ltd. https://www.tandfonline.com/doi/abs/10.1128/mcb.14.7.4912-4919.1994 (accessed on 28 January 2024).

**Figure 3 jof-10-00114-f003:**
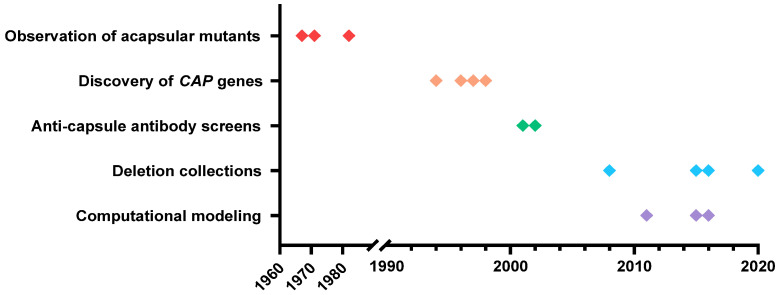
Timeline of major discoveries cited in the text.

## Data Availability

Not applicable.

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
