# Peer review of "A Quick reCAP: Discovering Cryptococcus neoformans Capsule Mutants"

_jof, 2024, doi:10.3390/jof10020114_

Round 1

Reviewer 1 Report

Comments and Suggestions for Authors

This opinion article from Boodwa-Ko and Doering provides a clear, brief summary of the different means by which genes involved in Cryptococcus capsule formation have been identified.  It will be a nice addition to the edition in honor of June Kwon-Chung.

Minor comments:  The last paragraph on page 4 describing the isolation of CAS1-6 via screening with a panel of antibodies is hard to follow and should probably be expanded a little to make it clearer.  Where does UXS1 come from?  Is it another name for one of the CAS genes?  Is it the case that CAS3 binds a new antibody (E1) that the parent strain does not?

Author Response

We appreciate the positive words of the reviewer. We have revised the indicated paragraph to address all three of these questions and to clarify our summary of these findings.

Reviewer 2 Report

Comments and Suggestions for Authors

This is a lovely and quite useful Short Review for the special issue to honor Dr. Kong-Chung's contribution to Mycology. I would only add the year of key discoveries in the text, considering that the citations are only numerical. Congratulations.

Author Response

We are grateful for the complementary comments of the reviewer. In response to their suggestion, we now refer to specific years in every main section of discoveries. We have also included an additional figure (Figure 3) which depicts a timeline of major discoveries cited in the text.

Reviewer 3 Report

Comments and Suggestions for Authors

The article “A quick reCAP: Discovering Cryptococcus neoformans Capsule Mutants” shows a review of the studies carried out by Dr. Kwon-Chung, which focused on the study of individual capsule-deficient mutants. Additionally, she discusses other approaches that contributed to the discovery of capsule mutants and enabled advancement in the field. It is an interesting and well-written work, however, I suggest that the authors include a flow chart, including the methods that have been used to evaluate capsule mutants, with the genes involved in capsule biosynthesis and regulation, which which would be very illustrative for readers.

Author Response

We thank the reviewer for their feedback. To address their suggestion and illustrate the progression of this field, we have included an additional figure (Figure 3) with a timeline of major discoveries cited in the text.
We would have liked to accommodate the suggestion to include the genes involved in capsule biosynthesis and regulation, but their sheer number make it impossible to concisely depict them in a figure. Additionally, the function of some of these genes, including the original CAP genes discovered by June Kwon-Chung, are still not known. Instead, we refer readers to reviews of this topic.